# Mathematical Model of Constitutive Relation and Failure Criteria of Plastic Concrete under True Triaxial Compressive Stress

**DOI:** 10.3390/ma14010102

**Published:** 2020-12-29

**Authors:** Liangming Hu, Shuyu Li, Junfu Zhu, Xu Yang

**Affiliations:** School of Water Conservancy Engineering, Zhengzhou University, Zhengzhou 450001, China; liangmingh@zzu.edu.cn (L.H.); yangtao199485@gs.zzu.edu.cn (J.Z.); vharp@gs.zzu.edu.cn (X.Y.)

**Keywords:** plastic concrete, constitutive relation, failure criteria, true triaxial compressive stress, stress-strain behavior

## Abstract

To establish the mathematic model of the constitutive relation and failure criteria of plastic concrete under true triaxial compressive stress, uniaxial compressive strength and true triaxial compressive strength of plastic concrete under three kinds of confining pressures with a size of 150 × 150 × 150 mm^3^ and a curing age of 540 days were tested, and the elastic modulus of plastic concrete with a size of 150 × 150 × 300 mm^3^ and a curing age of 90 days was tested. Based on the database, under uniaxial compressive stress tests and true triaxial compressive stress tests, the mathematic model of constitutive relation and the failure criteria of plastic concrete were investigated. It was observed that the strength of plastic concrete increased with confining stress. The mathematic model of constitutive relation in the form of the quartic polynomial is in good agreement with measured data. The general equations of failure criteria based on the octahedral stress-space under true triaxial compressive stress in the form of quadratic polynomial are well-fitting with experimental data. The mathematic model of constitutive relation and failure criteria of plastic concrete could provide the basis for a numerical simulation analysis of plastic concrete under true triaxial compressive stress, as well as promote the engineering application of plastic concrete.

## 1. Introduction

Plastic concrete is considered a five-phase construction material composed of cement, water, aggregate, and bentonite, which has great deformation capacity under load, excellent ductility after reaching failure, appropriate impermeability, and desirable shear strength [1,2,3]. The excellent deformation capacity of plastic concrete decreases both crack opening width and rupture probability [4]. Due to the great advantages of plastic concrete, it has been widely used worldwide in dam remediation and cut-off walls for many years [5].

There are considerable loads applied on the dam remediation and cut-off wall, therefore the research on mechanical properties and plastic concrete properties is of great significance. Experiments under confined and triaxial compressive stress have been conducted to investigate the properties of plastic concrete. The researches revealed that uniaxial compressive strength and modulus of elasticity of plastic concrete decreased by the increase of the bentonite or water-to-cement ratio [6,7]. The water-to-cement ratio of the plastic concrete was higher than standard concrete [8]. The internal friction angle increased with the increase of bentonite content. Bentonite enhanced the impermeability of plastic concrete. The failure strains of triaxial tests were shown to be three times that of failure strains obtained via biaxial tests [9,10]. Comparing to natural pozzolan and silica fume alone, certain natural pozzolan-silica fume combinations could better improve the strength, elastic modulus and workability of concrete [11]. The incorporation of 0.2% and 1%CNC could increase the compressive strength of cement by 10% and 17% respectively [12].

To investigate the mechanical behavior and hydraulic conductivity of plastic concrete, experimental tests have been done under various confinement conditions. In these studies, the effects of age, confining pressure, and mix proportions on the strength-deformation parameters and elastic deformability under uniaxial and multi-axial compressive stress have been assessed. Plastic concrete strength increased with age and cement strength class increasing [6,7]. With the use of fibers in plastic concrete, the strength and elastic modulus decreased and deformation increased [13]. The studies showed that with the increase of confining pressures applied on the specimens of plastic concrete under triaxial compressive stress, not only the compressive strength increased considerably but also the behavior of the plastic concrete to be alike the more ductile materials [14,15,16]. In the multi-axial tests, the application of confining compressive stress hindered the lateral strain under uniaxial compressive stress, thereby enhancing the compressive load-bearing capacity of plastic concrete specimens [17]. The stress–strain models and strength calculation model of plastic concrete under uniaxial and multi-axial compressive stress have been developed to predict the peak conditions and the complete stress–strain behavior of plastic concrete [18,19].

The failure criteria have been proposed to predict the mechanical properties of different combinations of concrete. The influence of silica fume and fly ash to self-compacting concrete under uniaxial and triaxial compressive stress and combined compression-shear performance has been explored. With the ash substitution rate increasing, the peak stress and peak strain of self-compacting concrete increased first and then decreased. The strength of self-compacting concrete increased with the addition of silica fume. Moreover, the failure criteria of self-compacting concrete based on the theory of Mohr–Coulomb and octahedral stress space have been proposed separately [20,21]. The constitutive models have been proposed to predict the mechanical properties of concrete under different temperatures and the effect of temperature [22].

The characteristics and failure criterion of the plastic concrete under true triaxial compressive stress were analyzed [23]. To date, due to the lack of substantiated scientific investigations and suitable constitutive laws, not least the experimental tests under true triaxial, plastic concrete is considered to be a linear-elastic material, which neglects the viscous behavior of plastic concrete during serviceability. Moreover, few studies have been sufficient enough to provide a reliable estimation of properties of plastic concrete under long-time load, so the constitutive law of plastic concrete could not be developed systematically [5]. In this article, the effect of confining compression on the strength of plastic concrete under true triaxial compressive stress was investigated with a curing age of 540 days. The mathematical model of constitutive relation, failure criteria, and the general equations of failure criteria under octahedral stress space of plastic concrete under true triaxial compressive stress with a curing age of 540 days were established.

## 2. Plastic Concrete Specimens Preparation

### 2.1. Materials and Specimens

Herein, 66 samples with a size of 150 × 150 × 150 mm^3^ and 33 samples with a size of 150 × 150 × 300 mm^3^ with 11 groups of mix proportions of plastic concrete were proportioned to study the constitutive model and failure criteria of plastic concrete. The mix proportions of plastic concrete specimens are reported in Table 1. Each group with a size of 150 × 150 × 150 mm^3^ and a curing age of 540 days had 6 samples that 3 samples were prepared to test the uniaxial compressive strength and 3 samples were proportioned to test true triaxial compressive strength. When the curing age is 540 days, the development of the strength of plastic concrete tends to be stable, and the influence of curing age on strength can be ignored. There were 3 samples in each group with a size of 150 × 150 × 300 mm^3^ and a curing age of 90 days prepared to test the measured elastic modulus under uniaxial compressive stress. According to the mix proportions in Table 1, it can be seen from Table 1 that WB100 and C150, S05 and C120, and B70 and CL180 are characterized by the proportions of the same ingredients. The data in Table 1 are arranged according to the change of a parameter.

The particle size distribution curves of the sand and gravel aggregates contained in the plastic concrete specimens are presented in Figure 1. The fitness modulus of medium sand was 2.7, coarse sand was 3.3. The cement incorporated in mix proportions was Type P.O 42.5 Portland cement [24]. Table 2 provides the mechanical properties of the used cement. The chemical properties of bentonite used for investigations are presented in Table 3. The silty clay was dried and ground to powder according to the particle size standard of 350 mesh. The mass of binder was the total mass of cement, bentonite and clay.

Using a forced mixer with appropriated rotation mixed the determined plastic concrete mix proportion until a homogeneous mixture was obtained. After mixing, the plastic concrete was placed in 150 × 150 × 150 mm^3^ steel molds and 150 × 150 × 300 mm^3^ respectively. After placing the specimens in wet rooms for 48 h, the samples were removed from the molds and kept curing in a standard curing room with the temperature range of 20 ± 3 °C and relative humidity of 95% for 540 days and 90 days respectively.

### 2.2. Triaxial Apparatus and Tests Procedure

The loading model of the monotonic triaxial compression test is applied confining loads first and then axial load. True triaxial compression test applies confining loads and axial load at the same time.

LY-C tension and compressive stress true triaxial apparatus was used in true triaxial compressive strength tests. The loads in three directions of apparatus are vertical-orthogonal and controlled independently. The maximum compressive load in the uniaxial of apparatus is 450 kN and the maximum tensile load is 75 kN with the load precision error less than 5%. LY-C tension and compressive stress true triaxial apparatus can carry out uniaxial, biaxial and triaxial compressive tests. Displacement meters were symmetrically arranged on the side of specimens. Displacement data were collected by the computer automatically.

A 3000 kN digital pressure testing machine was used in uniaxial compressive tests. Max capacity is 3000 kN and load precision error is less than 1%. A steel plate was adding to top of the specimens and the electronic displacement meter was fixed on the pressure plate under the testing machine. The data of axial load and deformation of specimens was collected through an automatic data acquisition instrument. The loading speed of specimens with the size of 150 × 150 × 150 mm^3^ was 0.1 MPa/s and the size of 150 × 150 × 300 mm^3^ was 0.005–0.01MPa/s.

Defining confining pressure values as *σ*_1_ and *σ*_2_, axial pressure value as *σ*_3_, and *σ*_1_ ≤ *σ*_2_ ≤ *σ*_3_, true triaxial compressive tests were performed with three confining pressures of (*σ*_1_ = 0.2 MPa, *σ*_2_ = 0.4 MPa), (*σ*_1_ = 0.4 MPa, *σ*_2_ = 0.6 MPa) and (*σ*_1_ = 0.4 MPa, *σ*_2_ = 0.8 MPa). Remained confining pressure unchanged after reaching the preset value, and axial pressure continues to increase until the specimen was damaged. The peak value of the stress-strain relation of *σ*_3_ was the strength of the specimen under true triaxial compressive stress. The test load was applied at a rate of 0.4 MPa per stage.

According to GB/T 50081-2002 [25], the strength of plastic concrete specimens under the uniaxial compressive stress of each group is the arithmetic mean value measured by 3 specimens (accurate to 0.1 MPa). If the difference between the uniaxial compressive stress of one specimen and the intermediate value exceeds 15% of the intermediate value, the uniaxial compressive strength of plastic concrete of this group takes the intermediate value. If the difference between both maximum value and minimum value with intermediate value all exceeds 15% of the intermediate value, the test results of this group of specimens are invalid.

## 3. Experimental Results and Analysis

### 3.1. Database of Uniaxial and True Triaxial Compressive Tests

Referring to Table 4, it is shown the elastic modulus of plastic concrete under uniaxial compressive stress with a curing age of 90 days and a size of 150 × 150 × 300 mm^3^, recording this elastic modulus as *E*_0_. The compressive strength of plastic concrete under uniaxial compressive stress, the compressive strength of plastic concrete under true triaxial compressive stress (*σ*_3_) and the percentage increase of true triaxial compressive strength relative to uniaxial compressive strength with a size of 150 × 150 × 150 mm^3^ and a curing age of 540 days are also presented in Table 4. The confining pressure of true triaxial tests is noted as (*σ*_1_, *σ*_2_). The confining pressure of true triaxial test is noted as (0.2 MPa, 0.4 MPa), (0.4 MPa, 0.6 MPa), (0.4 MPa, 0.8 MPa).

It is presented in Table 4 and Figure 2 that the strength of plastic concrete under true triaxial compressive tests is greater than uniaxial compressive tests. The strength of plastic concrete under true triaxial compressive tests increases with confining pressure. In true triaxial compressive tests, lateral strain under uniaxial compressive stress is hindered by the application of confining compressive stress, thereby enhancing the compressive load-bearing capacity of plastic concrete specimens.

The strength of plastic concrete decreases with the increase of water-to-binder ratio, and under different confining pressures and water-to-binder ratios, the triaxial compressive strength of plastic concrete increases by more than 114% compared with uniaxial compressive strength.

The effect of sand-to-total mass ratio on the strength of plastic concrete is not obvious. When the sand-to-total mass ratio is constant, the triaxial compressive strength of plastic concrete basically increases with the increase of confining pressure. When the sand-to-total mass ratio is 0.6, the triaxial compressive strength of plastic concrete increases by 181.85% compared with uniaxial compressive strength.

The true triaxial compressive strength of plastic concrete increases with the increase of cement mass. Under different confining pressures and cement content, the triaxial compressive strength of plastic concrete increases by more than 70% compared with uniaxial compressive strength.

The strength of plastic concrete decreases with the increase of clay mass. Under different confining pressures and clay content, the triaxial compressive strength of plastic concrete increases by more than 95% compared with uniaxial compressive strength.

The true triaxial compressive strength of plastic concrete decreases with the increase of bentonite mass. When bentonite mass is constant, the triaxial compressive strength of plastic concrete basically increases with the increase of confining pressure. Under different confining pressures and bentonite mass, the triaxial compressive strength of plastic concrete increases by more than 95% compared with uniaxial compressive strength.

### 3.2. Mathematical Model of Constitutive Relationship of Plastic Concrete under Triaxial Compressive Stress

At present, the mathematical model of the constitutive relationship of plastic concrete has mathematical models of linear-elastic, non-linear-elastic, and non-elastic constitutive relations. Plastic concrete is a typical nonlinear material. The mathematical model of linear-elastic constitutive relation cannot describe the constitutive relationship of plastic concrete accurately. The mathematical model of non-elastic constitutive relation is not intuitive in mathematical form, complicated in the derivation process, and not convenient for practical engineering. Therefore, this paper put forward a mathematical model of quartic polynomial conformed to the behavior of plastic concrete under true triaxial compressive stress.

#### 3.2.1. Calculation Formula of Peak Secant Modulus

Researches have been carried out that the elastic modulus of plastic concrete gradually increased with the increase of strength, but had great discreteness. The results of triaxial compressive tests carried out the relationship between secant elastic modulus and peak stress. Their relationship could be expressed by Equation (1). The value of *a*_1_ and *a*_2_ can be calculated by Equations (2) and (3) respectively.
(1)$$E_{f}=a_{1}\sigma_{3f}-a_{2}$$
(2)$$a_{1}=32.075\sigma_{1}-37.746\frac{\sigma_{1}}{\sigma_{2}}+55.522$$
(3)$$a_{2}=-333.45\sigma_{1}+387.36\frac{\sigma_{1}}{\sigma_{2}}-253.19$$
where *E_f_* is the peak secant elastic modulus of plastic concrete under triaxial pressure (MPa). $\sigma_{3f}$ is the compressive strength of plastic concrete under true triaxial tests (MPa). $\sigma_{1},\sigma_{2}$ is the confining compressive stress of true triaxial tests (MPa).

According to the empirical formula of elastic modulus under uniaxial compressive stress suggesting by ACI318-08 and GB50010-2002 [26,27]. Fitting the experimental data of plastic concrete in this experiment, the empirical formula of peak secant elastic modulus of plastic concrete under true triaxial compressive stress was obtained as Equations (4) and (7):(4)$$E_{f}=b_{1}\sqrt{\sigma_{3f}}+b_{2}$$

Parameters *b*_1_ and *b*_2_ can be calculated according to Equations (5) and (6).
(5)$$b_{1}=302.7\sigma_{1}-199.8\frac{\sigma_{1}}{\sigma_{2}}+283.69$$
(6)$$b_{2}=-947.25\sigma_{1}+641.04\frac{\sigma_{1}}{\sigma_{2}}-600.03$$
(7)$$E_{f}=\frac{10^{5}}{c_{1}+\frac{c_{2}}{\sigma_{3f}}}$$

Parameters *c*_1_ and *c*_2_ can be calculated according to Equations (8) and (9).
(8)$$c_{1}=1742.65\sigma_{1}-1074.72\frac{\sigma_{1}}{\sigma_{2}}+481.72$$
(9)$$c_{2}=-191.315\sigma_{1}+91.032\frac{\sigma_{1}}{\sigma_{2}}+4.052$$

The average values of the ratio of calculated values to test values of Equations (1), (4) and (7) are shown in Table 5.

From the results shown in Table 4, the average values of the ratio of calculated value to test the value of Equations (1), (4) and (7) are 1.0604, 1.0624 and 0.9992 respectively, so the fitting effect of Equation (7) is better than Equations (1) and (4). The peak secant modulus calculation formula adopted Equation (7).

#### 3.2.2. Mathematical Model of Constitutive Relation of Plastic Concrete under True Triaxial Compressive stress

According to data obtained from the true triaxial compressive stress tests of plastic concrete, a mathematical model of the quartic polynomial was established, which agreed well with the measured values.
(10)$$y=Ax^{4}+Bx^{3}+Cx^{2}+Dx,\;\mathrm{if}\;x\leq 1$$
(11)y=1, if x>1

The meanings of *x* and *y* in Equations (10) and (11) are showing in Equations (12) and (13).
(12)$$y=\frac{\sigma_{3}}{\sigma_{3f}}$$
(13)$$x=\frac{\varepsilon_{3}}{\varepsilon_{3f}}$$
where $\varepsilon_{3f}$ is the peak compressive strain.
(14)$$\mathrm{If}\;\varepsilon_{3}=\varepsilon_{3f},\;\sigma_{3}=\sigma_{3f}(x=1,\;y=1),\;A+B+C+D=1$$

Both sides of Equation (10) differentiated $\varepsilon_{3}$ simultaneously, getting Equation (15).
(15)$$\frac{dy}{d\varepsilon_{3}}=\left(4Ax^{3}+3Bx^{2}+2Cx+D\right)\frac{dx}{d\varepsilon_{3}}$$

The finishing equation is available shown as Equation (16).
(16)$$\frac{1}{\sigma_{3f}}\frac{d\sigma_{3}}{d\varepsilon_{3}}=\left(4Ax^{3}+3Bx^{2}+2Cx+D\right)\frac{1}{\varepsilon_{3f}}$$

Substituting $\varepsilon_{3}=0$(x=0) into Equation (16) can get Equation (17).
(17)$${\frac{d\sigma_{3}}{d\varepsilon_{3}}|}_{\varepsilon_{3}=0}=\frac{\sigma_{3f}}{\varepsilon_{3f}}D,\;D=\frac{E_{0}^{\prime }}{E_{f}}$$
where *E*_0_^′^ is the initial tangent elastic modulus of plastic concrete under true triaxial compressive stress. The expression of *E*_0_^′^ is shown in Equation (18).
(18)$$E_{0}^{\prime }={\frac{d\sigma_{3}}{d\varepsilon_{3}}|}_{\varepsilon_{3}=0}$$

It can be seen from Equation (17) that the coefficient *D* of the model is related to *E*_0_^′^/*E_f_*. By analogy, the coefficients *A*, *B*, *C*, *D* are related to *E*_0_^′^/*E_f_*. Because the initial tangent elastic modulus *E*_0_^′^ under triaxial compressive stress is difficult to measure in statistical analysis. Therefore, *E*_0_^′^ is replaced by elastic modulus (*E*_0_) measured under uniaxial compressive stress with a curing age of 90d.

Peak secant modulus *E_f_* calculated by Equation (7). The equation of each plastic concrete specimen is obtained by fitting the measured stress-strain curve, thus 33 *A*, *B*, *C*, *D* values are obtained. The expression of *A*, *B*, *C*, *D* can be obtained. The equations are shown in Equations (19)–(22).
(19)$$A=-0.1756{\left(\frac{E_{0}}{E_{f}}\right)}^{2}+1.6224\frac{E_{0}}{E_{f}}-0.1773$$
(20)$$B=0.1583{\left(\frac{E_{0}}{E_{f}}\right)}^{2}-1.0175\frac{E_{0}}{E_{f}}-7.0733$$
(21)$$C=0.1237{\left(\frac{E_{0}}{E_{f}}\right)}^{2}-1.8488\frac{E_{0}}{E_{f}}+11.294$$
(22)$$D=-0.1064{\left(\frac{E_{0}}{E_{f}}\right)}^{2}+1.2452\frac{E_{0}}{E_{f}}-3.0482$$
where *E*_0_ is the uniaxial compressive elastic modulus of plastic concrete with a curing age of 90 d.

#### 3.2.3. Comparison of Curves Obtained from Theoretical Model and Experiment Measured

The peak compressive strength obtained from true triaxial compressive tests under three confining stress is compared with the peak compressive strength obtained from Equations (10) and (11) to verify the reliability of the mathematical model. The results of peak compressive stress and the comparisons of calculated results and test results are shown in Table 6 and Figure 3, Figure 4 and Figure 5.

It can be seen from Table 6, Figure 3, Figure 4 and Figure 5, the peak compressive strength obtained from the theoretical model is in good agreement with the compressive stress obtained from experimental tests. The ensemble average value of the ratio of *σ*_3_/*σ*_3*f*_ is 1.0037, the standard deviation is 0.0031 and the coefficient of variation is 0.0030, which fully verifies the reliability of the established theoretical model of the quartic polynomial.

### 3.3. Failure Criteria of Plastic Concrete under True Triaxial Compressive Test

The plastic concrete failure criteria are processing a large number of triaxial test data of plastic concrete, drawing the failure envelope surface in principal stress space, and finding the appropriate mathematical expression according to the geometric characteristics of the envelope surface.

The strength in the triaxial space of each plastic concrete specimens (*f*_1_, *f*_2_, *f*_3_) obtained from the experiment is calibrated to the principal stress coordinate space (*σ*_1_, *σ*_2_, *σ*_3_), then connect the adjacent points with smooth curved surface to obtain failure envelop of plastic concrete.

There is a hydrostatic pressure axis in the coordinate space of principal stress, and the stress of each point on the axis is *σ*_1_= *σ*_2_= *σ*_3_. The distance between the point of the hydrostatic pressure axis and the origin of coordinates is hydrostatic pressure, with a value of $\xi =\sqrt{3}\sigma_{1}$. The angle between each principal stress axis and hydrostatic pressure axis is $\alpha =\arccos(1/\sqrt{3})$. The deviatoric plane is perpendicular to the hydrostatic axis. The first principal stress invariant (*I*_1_) is a constant which is the sum of the three principal stresses at each point on the same deflection plane [28].
(23)$$\sigma_{1}+\sigma_{2}+\sigma_{3}=I_{1}=\mathrm{Const}$$

Based on 33 test points of plastic concrete measured by true triaxial compressive stress tests and the general equation of failure criteria of octahedral space for ordinary concrete proposed by reference [29,30], this paper proposes a failure criterion in the form of the quadratic polynomial with the dimensionless expression of stress in octahedral space for plastic concrete. Its general expression is shown as Equation (24).
(24)$$\frac{\sigma_{\mathrm{oct}}}{f_{c}^{*}}=a(\frac{\tau_{\mathrm{oct}}}{f_{c}^{*}}{)}^{2}+b(\frac{\tau_{\mathrm{oct}}}{f_{c}^{*}})+c$$

The parameters in Equation (24) calculate according to the following equations.
(25)$$b=b_{t}{\left(\cos\frac{3}{2}\theta \right)}^{1.5}+b_{c}{\left(\sin\frac{3}{2}\theta \right)}^{2}$$
(26)$$\sigma_{\mathrm{oct}}=\frac{1}{3}(f_{1}+f_{2}+f_{3})$$
(27)$$\tau_{\mathrm{oct}}=\frac{1}{3}\sqrt{{\left(f_{1}-f_{2}\right)}^{2}+{\left(f_{2}-f_{3}\right)}^{2}+{\left(f_{3}-f_{1}\right)}^{2}}$$
(28)$$\theta ={\mathrm{arc}}_{}^{}cos(\frac{2f_{1}-f_{2}-f_{3}}{3\sqrt{2}\tau_{\mathrm{oct}}})$$
where $\sigma_{\mathrm{oct}}$ is the normal stress of octahedral plastic concrete. $f_{c}^{*}$ is the uniaxial compressive strength of plastic concrete at an age of 540 days. *θ* is the included angle of the offset plane. $\tau_{\mathrm{oct}}$ is octahedral plastic concrete shear stress.

The values of parameters *a*, *b*(*b*_t_, *b*_c_), c can be determined by experiment.

Define γ as the relative normal stress of the octahedral plastic concrete, χ as the relative shear stress of the octahedral plastic concrete, the expressions of *γ* and *χ* are in Equations (29) and (30).
(29)$$\gamma =\frac{\sigma_{oct}}{f_{c}^{*}}$$
(30)$$\chi =\frac{\tau_{oct}}{f_{c}^{*}}$$

Equation (31) is the general expression of Equation (24).
(31)$$\gamma =a\chi^{2}+b\chi +c$$

In the experiment of plastic concrete under true triaxial pressure, octahedral normal stress, shear stress and Lode’s angle in the stress space are shown in Table 7, Table 8 and Table 9 when three confining pressures are (0.2q, 0.4q), (0.4q, 0.6q), (0.4q, 0.8q) respectively.

Based on the data of true triaxial compressive tests, the general equation of failure criteria has been studied as the expression of Equation (32).
(32)$$\gamma =-0.033\chi^{2}+0.896\chi -0.038$$

A comparison between the measured values and the calculated values of Equation (32) is shown in Figure 6. The average value of the ratio of the calculated value to the experimental value is 1.002. The coefficient of variation is 0.0444. The standard deviation is 0.0445. It can be seen that the suggested failure criteria equation is in good agreement with the experimental data.

## 4. Conclusions

In this article, the mechanical performance of plastic concrete was tested and theoretically analyzed. Firstly, the results of the strength of plastic concrete under true triaxial and uniaxial compressive stress were obtained through experiments. Secondly, according to the results of the tests, the relationships between materials and confining pressure on strength of plastic concrete were analyzed. Thirdly, the mathematic models of the constitutive relationship and failure criteria based on the strength of plastic concrete under true triaxial tests were established. The following conclusions can be obtained:(1)According to the data shown in Table 4, the strength of plastic concrete under true triaxial compressive tests was greater than strength under uniaxial compressive tests due to lateral strain prevented by confining compressive stress. The triaxial compressive strength of plastic concrete increased by more than 70% compared with uniaxial compressive strength. The maximum growth rate could reach 181.85%.(2)With the increase of confining compressive stress, the lateral compressive stress had a better restraining effect on the transverse deformation of plastic concrete specimens, which delayed the appearance and development of micro-cracks in plastic concrete specimens. Therefore, the strength of plastic concrete increased with the confining compressive stress.(3)Based on the experimental data and the existing constitutive model of concrete under triaxial compressive stress, a quartic polynomial constitutive model conforming to the constitutive characteristics of plastic concrete was established for the first time. Compared with the measured data, the ensemble average value of the ratio of *σ*_3_/*σ*_3f_ was 1.0037, the standard deviation was 0.0031, and the coefficient of variation was 0.0030. So, the results of the mathematical model were in good agreement with the data obtained from the experiment.(4)The failure criteria of plastic concrete under triaxial compressive stress in octahedral stress space was established. The general equation of the failure criteria was established according to the failure criteria. The average value of the ratio of calculated value to the experimental value was 1.002, the coefficient of variation was 0.0444, and the standard deviation was 0.0445, which proved that the calculated values of the failure criteria equation were in good agreement with the experimental values.

## Figures and Tables

**Figure 1 materials-14-00102-f001:**
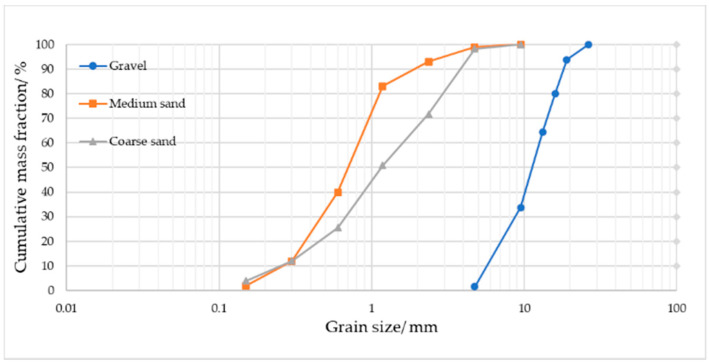
Particle size distribution curves for the sand and gravel aggregates.

**Figure 2 materials-14-00102-f002:**
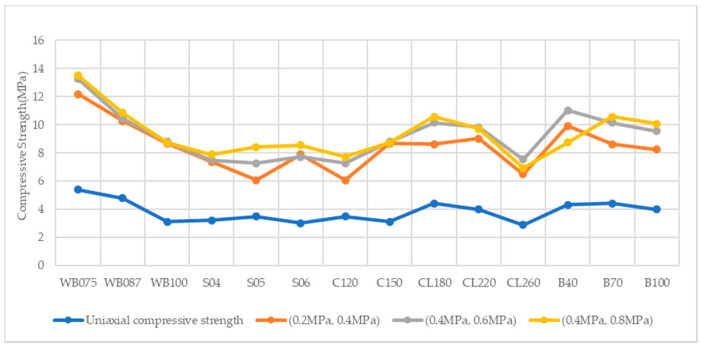
Compressive Strength under Different Confining Compression.

**Figure 3 materials-14-00102-f003:**
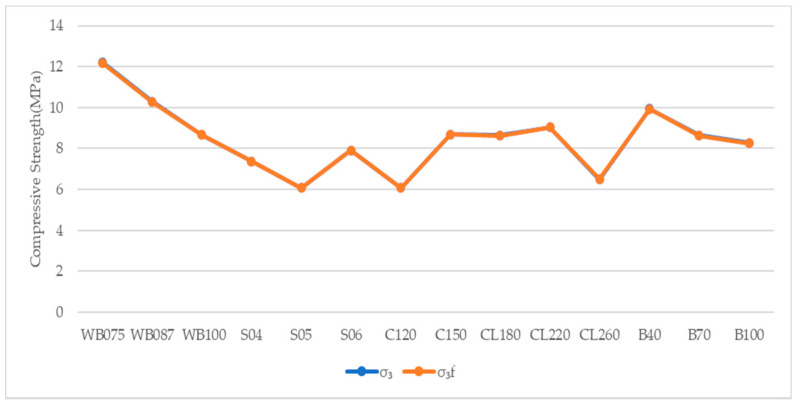
Comparison of Peak Compressive Strength between Calculation and Test with Confining pressure (0.2 MPa, 0.4 MPa).

**Figure 4 materials-14-00102-f004:**
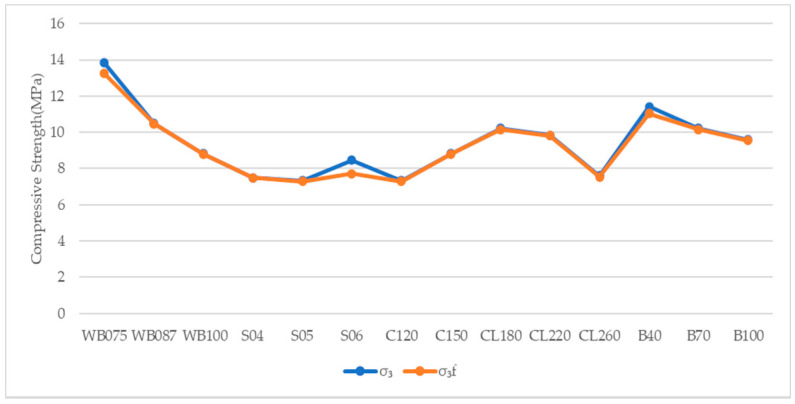
Comparison of Peak Compressive Strength between Calculation and Test with Confining pressure (0.4 MPa, 0.6 MPa).

**Figure 5 materials-14-00102-f005:**
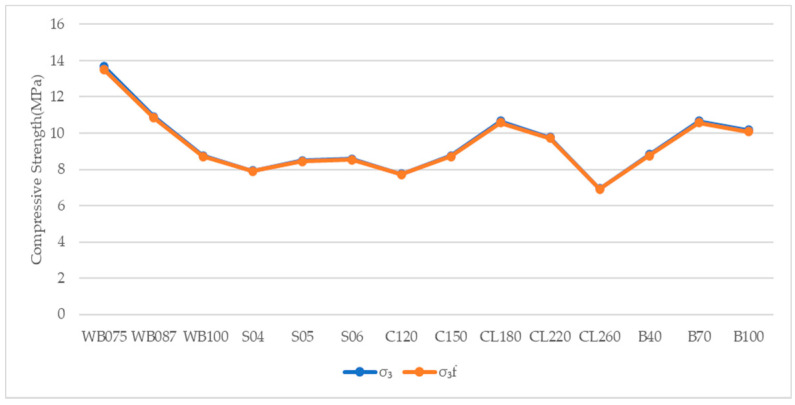
Comparison of Peak Compressive Strength between Calculation and Test with Confining pressure (0.4 MPa, 0.8 MPa).

**Figure 6 materials-14-00102-f006:**
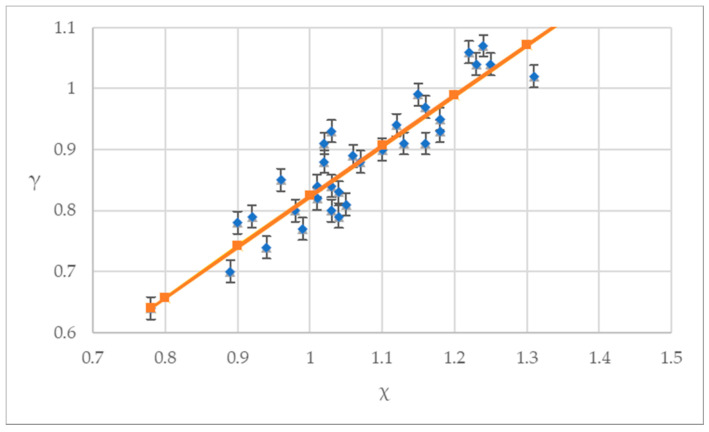
General equation curve and test value of plastic concrete failure criteria.

**Table 1 materials-14-00102-t001:** Test Mix Proportion Scheme.

Mix ID	Water-Binder Ratio	Sand Ratio	Water/(kg·m^−3^)	Cement/(kg·m^−3^)	Clay/(kg·m^−3^)	Bentonite/(kg·m^−3^)	Sand/(kg·m^−3^)	Gravel/(kg·m^−3^)
WB075	0.75	0.50	307.50	150	180	80	666.25	666.25
WB087	0.87	0.50	356.70	150	180	80	641.65	641.65
WB100	1.00	0.50	400	150	180	70	625	625
S04	1.00	0.40	370	120	180	70	524	786
S05	1.00	0.50	370	120	180	70	655	655
S06	1.00	0.60	370	120	180	70	786	524
C120	1.00	0.50	370	120	180	70	655	655
C150	1.00	0.50	400	150	180	70	625	625
CL180	0.87	0.50	322	120	180	70	655	655
CL220	0.87	0.50	357	120	220	70	641.5	641.5
CL260	0.87	0.50	391	120	260	70	604.5	604.5
B40	0.87	0.50	296	120	180	40	707	707
B70	0.87	0.50	322	120	180	70	655	655
B100	0.87	0.50	348	120	180	100	651	651

Note: WB refers to the water-to-binder ratio. S refers to the sand-to-total mass ratio. C refers to cement. CL refers to clay. B refers to bentonite.

**Table 2 materials-14-00102-t002:** Properties of cement.

	Rupture Strength (MPa)		Compressive Strength (MPa)		Setting Time (Minutes)	
	3 d	28 d	3 d	28 d	Initial Setting Time	Final Setting Time
**Test value**	5.8	9.1	24.8	48.3	115	235

**Table 3 materials-14-00102-t003:** Chemical properties of bentonite.

Oxide	SiO_2_	Al_2_O_3_	Fe_2_O_3_	FeO	CaO	MgO	K_2_O	Na_2_O	MnO	TiO_2_	LOS
(%)	67.78	15.01	2.03	0.08	2.17	3.39	0.68	0.65	0.016	0.087	8.20

**Table 4 materials-14-00102-t004:** Summary of the results of the uniaxial compressive and true triaxial compressive tests.

Numbered	*E*_0_/(MPa)	Uniaxial Compressive Strength/(MPa)	Compressive Strength and Percentage Increase under True Triaxial Test/(MPa)					
			(0.2 MPa, 0.4 MPa)		(0.4 MPa, 0.6 MPa)		(0.4 MPa, 0.8 MPa)	
WB075	2078.17	5.40	12.18	125.56%	13.25	145.37%	13.50	150.00%
WB087	1372.60	4.79	10.28	114.61%	10.46	118.37%	10.85	126.51%
WB100	1328.37	3.12	8.67	177.88%	8.79	181.73%	8.71	179.17%
S04	1049.60	3.21	7.38	129.91%	7.49	133.33%	7.90	146.11%
S05	1308.80	3.50	6.07	73.43%	7.28	108.00%	8.43	140.86%
S06	1229.17	3.03	7.90	160.73%	7.71	154.46%	8.54	181.85%
C120	1308.80	3.50	6.07	73.43%	7.28	108.00%	7.71	120.29%
C150	1328.37	3.13	8.68	177.32%	8.79	180.83%	8.71	178.27%
CL180	1809.40	4.42	8.63	95.25%	10.16	129.86%	10.57	139.14%
CL220	1150.63	3.99	9.03	126.32%	9.82	146.12%	9.72	143.61%
CL260	992.52	2.90	6.50	124.14%	7.54	160.00%	6.91	138.28%
B40	1554.10	4.33	9.93	129.33%	11.03	154.73%	8.75	102.08%
B70	1809.40	4.42	8.63	95.25%	10.16	129.86%	10.57	139.14%
B100	1805.10	4.00	8.25	106.25%	9.54	138.50%	10.07	151.75%

**Table 5 materials-14-00102-t005:** Average Values of the Ratio of Calculated Value to Test Value.

Equation Number	Average Value
(1)	1.0604
(4)	1.0624
(7)	0.9992

**Table 6 materials-14-00102-t006:** Comparison of Peak Compressive Strength between Calculation and Test.

Numbered	Calculated Peak Compressive Strength/(MPa)						Total Calculated Value
	(0.2 MPa, 0.4 MPa)		(0.4 MPa, 0.6 MPa)		(0.4 MPa, 0.8 MPa)		
	*σ* _3_	*σ*_3_/*σ_3f_*	*σ* _3_	*σ*_3_/*σ_3f_*	*σ* _3_	*σ*_3_/*σ_3f_*	
WB075	12.2182	1.0031	13.8353	1.0074	13.6618	1.0120	
WB087	10.2864	1.0006	10.4959	1.0032	10.9173	1.0062	
WB100	8.6742	1.0005	8.8178	1.0030	8.7600	1.0057	
S04	7.3756	0.9994	7.5011	1.0013	7.9275	1.0035	
S05	6.0730	1.0005	7.3422	1.0028	8.4769	1.0056	
S06	7.9009	1.0001	8.4516	1.0023	8.5822	1.0049	
C120	6.0730	1.0005	7.3422	1.0028	7.7525	1.0055	
C150	8.6842	1.0005	8.8178	1.0030	8.7600	1.0057	
CL180	8.6507	1.0024	10.2214	1.0058	10.6724	1.0097	
CL220	9.0279	0.9998	9.84056	1.0019	9.7626	1.0044	
CL260	6.4948	0.9992	7.5987	1.0010	6.9305	1.0030	
B40	9.9434	1.0013	11.4214	1.0043	8.8159	1.0075	
B70	8.6507	1.0024	10.2214	1.0058	10.6724	1.0097	
B100	8.2698	1.0023	9.5974	1.0058	10.1669	1.0096	
average value		1.0009		1.0036		1.0066	1.0037
standard deviation		0.0012		0.0019		0.0026	0.0031
coefficient of variation		0.0012		0.0019		0.0026	0.0030

**Table 7 materials-14-00102-t007:** Normal stress, shear stress and Lode’s angle in octahedral stress space under (0.2 MPa, 0.4 MPa).

Test ID	*f* _1_	*f* _2_	*f* _3_	*σ_oct_*	*τ_oct_*	*θ*	*f_c_**	*γ*	*χ*
WB075	0.2	0.4	12.18	4.26	5.60	0.84	5.40	0.79	1.04
WB087	0.2	0.4	10.41	3.67	4.77	0.98	4.79	0.77	0.99
WB100, C150	0.2	0.4	8.97	3.19	4.09	1.14	3.13	1.02	1.31
S04	0.2	0.4	7.38	2.66	3.34	1.40	3.21	0.83	1.04
S05, C120	0.2	0.4	6.07	2.22	2.72	1.72	3.50	0.64	0.78
S06	0.2	0.4	7.90	2.83	3.58	1.31	3.03	0.93	1.18
CL180, B70	0.2	0.4	8.63	3.08	3.93	1.19	4.43	0.70	0.89
CL220	0.2	0.4	9.04	3.21	4.12	1.14	3.99	0.80	1.03
CL260	0.2	0.4	6.50	2.37	2.92	1.60	2.90	0.82	1.01
B40	0.2	0.4	9.93	3.51	4.54	1.03	4.33	0.81	1.05
B100	0.2	0.4	8.25	2.95	3.75	1.25	4.00	0.74	0.94

**Table 8 materials-14-00102-t008:** Normal stress, shear stress and Lode’s angle in octahedral stress space under (0.4 MPa, 0.6 MPa).

Test ID	*f* _1_	*f* _2_	*f* _3_	*σ_oct_*	*τ_oct_*	*θ*	*f_c_**	*γ*	*χ*
WB075	0.4	0.6	13.73	4.91	6.24	0.75	5.40	0.91	1.16
WB087	0.4	0.6	10.46	3.82	4.69	1.00	4.79	0.80	0.98
WB100, C150	0.4	0.6	8.79	3.26	3.91	1.20	3.13	1.04	1.25
S04	0.4	0.6	7.49	2.83	3.29	1.42	3.21	0.88	1.02
S05, C120	0.4	0.6	7.32	2.77	3.22	1.45	3.50	0.79	0.92
S06	0.4	0.6	8.43	3.14	3.74	1.25	3.03	1.04	1.23
CL180, B70	0.4	0.6	10.16	3.72	4.56	1.03	4.43	0.84	1.03
CL220	0.4	0.6	9.82	3.61	4.40	1.06	3.99	0.90	1.10
CL260	0.4	0.6	7.59	2.86	3.34	1.40	2.90	0.99	1.15
B40	0.4	0.6	11.37	4.12	5.12	0.91	4.33	0.95	1.18
B100	0.4	0.6	9.54	3.51	4.26	1.10	4.00	0.88	1.07

**Table 9 materials-14-00102-t009:** Normal stress, shear stress and Lode’s angle in octahedral stress space under (0.4 MPa, 0.8 MPa).

Test ID	*f* _1_	*f* _2_	*f* _3_	*σ_oct_*	*τ_oct_*	*θ*	*f_c_**	*γ*	*χ*
WB075	0.4	0.8	13.50	4.90	6.08	1.54	5.40	0.91	1.13
WB087	0.4	0.8	10.85	4.02	4.84	1.94	4.79	0.84	1.01
WB100, C150	0.4	0.8	8.72	3.31	3.83	2.44	3.13	1.06	1.22
S04	0.4	0.8	7.54	2.91	3.28	2.86	3.21	0.91	1.02
S05, C120	0.4	0.8	7.71	2.97	3.36	2.79	3.50	0.85	0.96
S06	0.4	0.8	8.54	3.25	3.75	2.50	3.03	1.07	1.24
CL180, B70	0.4	0.8	10.57	3.92	4.70	1.99	4.43	0.89	1.06
CL220	0.4	0.8	10.40	3.87	4.62	2.02	3.99	0.97	1.16
CL260	0.4	0.8	6.93	2.71	2.99	3.13	2.90	0.93	1.03
B40	0.4	0.8	8.87	3.36	3.90	2.40	4.33	0.78	0.90
B100	0.4	0.8	10.07	3.76	4.47	2.09	4.00	0.94	1.12

## Data Availability

The data used to support the findings of this study are included within the article.

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
