# Peer review of "Mathematical Model of Constitutive Relation and Failure Criteria of Plastic Concrete under True Triaxial Compressive Stress"

_materials, 2020, doi:10.3390/ma14010102_

Round 1
Reviewer 1 Report
Dear Editor,
The topic of the paper is interesting and suits the Journal of MDPI Materials. However, a major revision is required before this manuscript is qualified to be published in this prestigious journal. The manuscript is needed to be revised grammatically. The authors are required to check the whole manuscript with a grammar specialist as it has several grammatical errors. Only after revising the manuscript based on the comments, the paper is suggested to be published in MDPI. Further information on various issues identified in the manuscript appears below:
- The introduction section needs to be revised. A paragraph should be dedicated to the importance of your work.
- The authors have done a great job in the literature review. However, the introduction needs more attention. More information on the topic of this paper can be found here:
Ghahari, S.A. et al. "Fracture Properties Evaluation of Cellulose Nanocrystals Cement Paste." Materials 13, no. 11 (2020): 2507.
- Please provide more detailed reasoning behind the behavior. The details should include the rigid numbers or percentages.
- The theoretical analysis needs a more in-depth discussion.
- Please indicate how many samples for each experiment have been used. Please revise the other experiments respectively.
- Please describe the process of each experiment. Also indicate the model of each tool that is used in the experiment. What is the accuracy of each machine? Please explain them accurately.
- The conclusion needs more elaboration. Please use more sentences containing percentages and illustrate the main conclusions in the manuscript. Please paraphrase your results and discussions and use them in the conclusion part.
Author Response
Reviewer 1
Point 1.
The introduction section needs to be revised. A paragraph should be dedicated to the importance of your work.
Response 1:
I revised the introduction and added a paragraph of the importance of my work in line 69-78:
The characteristics and failure criterion of the plastic concrete under true triaxial compressive stress were analyzed. To date, due to the lack of substantiated scientific investigations and suitable constitutive laws, not least the experimental tests under true triaxial, plastic concrete is considered to be a linear-elastic materials, which neglects the viscous behavior of plastic concrete during serviceability. And few studies have been sufficient enough to provide reliable estimation of properties of plastic concrete under long-time load, so the constitutive law of plastic concrete could not be developed systematically. In this article, the effect of confining compression on the strength of plastic concrete under true triaxial compressive stress was investigated with a curing age of 540 days. The mathematical model of constitutive relation, failure criteria and the general equations of failure criteria under octahedral stress space of plastic concrete under true triaxial compressive stress with a curing age of 540 days were established.
Point 2.
The authors have done a great job in the literature review. However, the introduction needs more attention. More information on the topic of this paper can be found here:
Ghahari, S.A. et al. "Fracture Properties Evaluation of Cellulose Nanocrystals Cement Paste." Materials 13, no. 11 (2020): 2507.
Response 2:
I have added the article you recommended as reference 12.
Point 3.
Please provide more detailed reasoning behind the behavior. The details should include the rigid numbers or percentages.
Response 3:
I have added more detailed reasoning behind the behavior which includes the rigid numbers or percentages in line 162-180:
The strength of plastic concrete decreases with the increase of water-to-binder ratio, and under different confining pressures and water-to-binder ratios, the triaxial compressive strength of plastic concrete increases by more than 114% compared with uniaxial compressive strength.
The effect of sand-to-total mass ratio to strength of plastic concrete is not obvious. When the sand-to-total mass ratio is constant, the triaxial compressive strength of plastic concrete basically increases with the increase of confining pressure. When the sand-to-total mass ratio is 0.6, the triaxial compressive strength of plastic concrete increases by 181.85% compared with uniaxial compressive strength.
The true triaxial compressive strength of plastic concrete increases with the increase of cement content. Under different confining pressures and cement content, the triaxial compressive strength of plastic concrete increases by more than 70% compared with uniaxial compressive strength.
The strength of plastic concrete decreases with the increase of clay content. Under different confining pressures and clay content, the triaxial compressive strength of plastic concrete increases by more than 95% compared with uniaxial compressive strength.
The true triaxial compressive strength of plastic concrete decreases with the increase of bentonite content. When bentonite content is constant, the triaxial compressive strength of plastic concrete basically increases with the increase of confining pressure. Under different confining pressures and bentonite content, the triaxial compressive strength of plastic concrete increases by more than 95% compared with uniaxial compressive strength.
Point 4.
The theoretical analysis needs a more in-depth discussion.
Response 4:
I discuss the influence of different materials on strength of plastic concrete and the influence of confining pressure under different materials according to Table 4.
I added the figures of Table 6 as Figure 3, 4 and 5 to prove the accuracy of the formula. And I used more rigid numbers to prove the accuracy of the mathematic model in line 249-250.
Point 5.
Please indicate how many samples for each experiment have been used. Please revise the other experiments respectively.
Response 5:
I'm sorry that what I wrote about this part is not clear enough. I have supplemented this part. The number of samples for each experiment is shown in line 74-81:
Herein, 66 samples with a size of 150×150×150 mm3 and 33 samples with a size of 150×150×300 mm3 with 11 groups of mix proportions of plastic concrete have been proportioned to study the constitutive model and failure criteria of plastic concrete. The mix proportions of plastic concrete specimens are reported in Table 1. Each group with a size of 150×150×150 mm3 has 6 samples that 3 samples were prepared to test the strength of plastic concrete under uniaxial compression and 3 samples were proportioned to true triaxial test with curing age of 540 days. There are 3 samples in each group with a size of 150×150×300 mm3 were prepared to test the measured elastic modulus under uniaxial compressive stress with a curing age of 90 days.
Point 6.
Please describe the process of each experiment. Also indicate the model of each tool that is used in the experiment. What is the accuracy of each machine? Please explain them accurately.
Response 6:
The process of uniaxial compression experiment is shown in line 127-131:
Steel plate was adding on the top of test specimens and electronic displacement meter was fixed on the pressure plate under the testing machine. The data of axial load and deformation of the test specimens was collected through automatic data acquisition instrument. The loading speed of specimens with the size of 150×150×150 mm3 was 0.1 MPa/s and the size of 150×150×300 mm3 is 0.005-0.01MPa/s
The process of true triaxial compression test is written in line 122-127:
Defining confining pressures as σ1 and σ2, axial pressure as σ3, and σ1≤σ2≤σ3, true triaxial tests were performed with three confining pressure of (σ1= 0.2 MPa,σ2= 0.4 MPa), (σ1 =0.4 MPa, σ2=0.6 MPa) and (σ1=0.4 MPa, σ2=0.8 MPa). Remained confining pressure unchanged after reaching the preset value, and axial pressure continue to increase until the specimen is damaged. The peak value of stress-strain relation of σ3 is the strength under true triaxial compressive stress of specimen. Test load was applied at a rate of 0.4 MPa per stage.
The model of each tool that is used in the experiment and the accuracy of each machine is shown in line 109-118:
LY-C tension and compressive stress true triaxial apparatus was used in true triaxial compressive tests. The loads in three directions of apparatus are vertical-orthogonal and controlled independently. The maximum compressive load in uniaxial of apparatus is 450 kN and the maximum tensile load is 75 kN with the load precision error less than 5%. LY-C tension and compressive stress true triaxial apparatus can carry out uniaxial, biaxial and triaxial compressive tests. Displacement meters were symmetrically arranged on the side of specimens. Displacement data was collected by computer automatically.
3000 kN digital pressure testing machine is used in uniaxial compressive tests. Max capacity is 3000 kN, load precision is less than 1%. Steel plate was adding on the top of test specimens and electronic displacement meter was fixed on the pressure plate under the testing machine.
Point 7.
The conclusion needs more elaboration. Please use more sentences containing percentages and illustrate the main conclusions in the manuscript. Please paraphrase your results and discussions and use them in the conclusion part
Response 7:
I used more percentages to paraphrase in line 303-305, 312-315, 318-320:
The triaxial compressive strength of plastic concrete increased by more than 70% compared with uniaxial compressive strength. The maximum growth rate could reach 181.85%.
Compared with the measured data, the ensemble average value of the ratio of σ3 /σ3f was 1.0037, the standard deviation was 0.0031 and the coefficient of variation was 0.0030, so the results of the mathematical model were in good agreement with the data obtained from the experiment.
The average value of the ratio of calculated value to the experimental value was 1.002, the coefficient of variation was 0.0444, and the standard deviation was 0.0445 which proved that calculated values of failure criteria equation were in good agreement with the experimental values.
And I illustrated the main conclusion on line 295-300:
In this article, the mechanical performance of plastic concrete was tested and theoretically analyzed. Firstly, the strength of plastic concrete under true triaxial and uniaxial compressive stress were obtained through experiments. Secondly, according to results of the tests, the relationships between materials and confining pressure on strength of plastic concrete were analyzed. Thirdly, mathematic model of constitutive relationship and failure criteria based on strength of plastic concrete under true triaxial tests were established.

Reviewer 2 Report
Congratulations to authors. Interesting paper regarding a mathematic model of constitutive relation and failure criteria of plastic concrete under true triaxial compressive stress.
In general, the paper is well presented but some details shall be clarified:
- Abstract, Section 2.1 and others: please change “… 150×150×150 mm …” by “… 150×150×150 mm3 …” or “… 150 mm × 150 mm × 150 mm …”;
- Section 3.1: “… with curing age of 90 days …”, what kind of cure was used?
- Table (5): where did Authors write “Equation number (1), (2), (3)” wanted to write “Equation number (1), (4), (7)”?
- After Equation (22): please cut “:” in “… age of 90d.:”;
- Equation (28): please include a space in “Ï´=arc cos (…)”;
- After Equation (28): the paragraph “τoct is octahedral plastic concrete shear stress” is repeated;
- Tables 4 & 6 to 8: maybe one or two graphics can give a more interesting visual idea of the differences in some variables.
Author Response
Reviewer 2
Point 1.
Abstract, Section 2.1 and others: please change “… 150×150×150 mm …” by “… 150×150×150 mm3 …” or “… 150 mm × 150 mm × 150 mm …”
Response 1:
The “150×150×150 mm” in article has been changed by “150×150×150 mm3”.
Point 2.
Section 3.1: “… with curing age of 90 days …”, what kind of cure was used?
Response 2:
The kind of curing has shown in line 103-106 as :
After mixing, the plastic concrete was placed in 150×150×150 mm3 steel molds and 150×150×300 mm3 respectively. After placing the specimens in wet rooms for 48 hours, the samples were removed from the molds and kept curing in standard curing room with the temperature range of 20±3 ℃ and relative humidity of 95% for 540 days and 90 days respectively.
Point 3.
Table (5): where did Authors write “Equation number (1), (2), (3)” wanted to write “Equation number (1), (4), (7)”?
Response 3:
This is my fault. I have corrected the equation numbers in Table (5).
Point 4.
After Equation (22): please cut “:” in “… age of 90d.:”;
Response 4:
I have cut “:” in “… age of 90d.:”
Point 5.
Equation (28): please include a space in “Ï´=arc cos (…)”;
Response 5:
I have included a space in Equation 28. Please see the attachment .
Point 6.
After Equation (28): the paragraph “τoct is octahedral plastic concrete shear stress” is repeated;
Response 6:
I have deleted the repeated sentences.
Point 7.
Tables 4 & 6 to 8: maybe one or two graphics can give a more interesting visual idea of the differences in some variables
Response 7:
Dear Reviewer, according to your advice, I have added a figure under Table 4。
The figures of Table 4 and Table6 please see the attachment.
And the Figure 6 in the article shows the measured values shown in Table 7-9, the calculated values of Eq. (32) and the error bars between them.

Reviewer 3 Report
Undoubtedly, the original contribution of the article are the triaxial tests of plastic concrete. The use of such concretes in hydrotechnical construction engineering is becoming a must, and the triaxial stress state in massive structures such as dams is particularly frequent.
The introduction is a sufficient basis for further analyzes. Unfortunately, the description of the research itself is not entirely clear to me. I will start my review with general remarks.
The abstract does not fully describe the content of the article. Two types of samples (150 and 300 mm long) were tested, most after 90 days.
There is no explanation of the motivation why the samples were tested after 90 and 540 days?
I do not understand why two types of samples were tested (cubes 150x150x150 and prisms 150x150x300) - due to the possibility of determining the elastic modulus on prismatic samples?
Why do the authors use the unit q = 1MPa, when instead of 0.4q you can directly write 0.4MPa. At first I thought 0.4q meant that σ1 = 0.4 σ 3. Only the analysis of Table 7 showed that these are constant values.
In my opinion, the caption of Table 5 is not clear, the table does not compare the modules but their ratios.
I cannot find where the test results after 540 days were used. Theoretically, according to given in the text explanation of f*c, these are tables 7-9, but when analyzing f *c, I do not see any differences with Table 4, which shows the uniaxial compressive strength tests after 90 days.
In the last paragraph of p.2.2 you wrote when the results were rejected (15% deviation from the mean). Has this situation occurred and which models does it apply to?
I am not a native speaker, but I can see some significant language mistakes. The text should be checked.
Other more detailed comments:
Caption of Tables 7-9 - I think it's about Lode's angle, not Rhode
Figure 2 may be slightly enlarged.
Point 4 – Conclusions (there is more than one conclusion)
Due to the indicated shortcomings, I recommend a major revision of the manuscript.
Author Response
Reviewer 3
Point 1.
The abstract does not fully describe the content of the article. Two types of samples (150 and 300 mm long) were tested, most after 90 days.
Response 1 :
I have described the content of article in detail in abstract according to your opinion. The detailed description is shown in line 11-14:
To establish the mathematic model of constitutive relation and failure criteria of plastic concrete under true triaxial compressive stress, uniaxial compressive strength and true triaxial compressive strength of plastic concrete under 3 kinds of confining pressures with a size of 150×150×150 mm3 and a curing age of 540 days were tested, and elastic modulus of plastic concrete with a size of 150×150×300 mm3 and a curing age of 90 days were tested.
Point 2.
There is no explanation of the motivation why the samples were tested after 90 and 540 days?
Response 2 :
The explanation of motivation that the samples were tested after 540 days I have added in line 87-89:
When the curing age is 540 days, the development of strength of plastic concrete tends to be stable, and the influence of curing age on strength can be ignored.
The explanation of motivation that the samples were tested after 90 days I have explained it in line 89-91 and line 225-226:
There were 3 samples in each group with a size of 150×150×300 mm3 and a curing age of 90 days prepared to test the measured elastic modulus under uniaxial compressive stress.
Because the initial tangent elastic modulus E0’ under triaxial compressive stress is difficult to measure in statistical analysis. Therefore, E0’ is replaced by elastic modulus (E0) measured under uniaxial compressive stress with a curing age of 90d.
Point 3.
I do not understand why two types of samples were tested (cubes 150x150x150 and prisms 150x150x300) - due to the possibility of determining the elastic modulus on prismatic samples?
Response 3 :
The reason of two types of sample were tested is that cubes150×150×300 mm3 were prepared to test the elastic modulus under uniaxial compressive stress with a curing age of 90 days. The reason is written in line 89-91:
There were 3 samples in each group with a size of 150×150×300 mm3 and a curing age of 90 days prepared to test the measured elastic modulus under uniaxial compressive stress.
Point 4.
Why do the authors use the unit q = 1MPa, when instead of 0.4q you can directly write 0.4MPa. At first I thought 0.4q meant that σ1 = 0.4 σ 3. Only the analysis of Table 7 showed that these are constant values.
Response 4 :
I am sorry that my expression in the article make you misunderstanding. I have changed 0.2q, 0.4q, 0.8q to 0.2MPa, 0.4MPa, 0.8MPa in the article.
Point 5.
In my opinion, the caption of Table 5 is not clear, the table does not compare the modules but their ratios.
Response 5:
I have changed the caption of Table 5 “Comparison of Elastic Modulus Calculation and Test” to “Average Values of the Ratio of Calculated Value to Test Value”.
Point 6.
I cannot find where the test results after 540 days were used. Theoretically, according to given in the text explanation of f*c, these are tables 7-9, but when analyzing f *c, I do not see any differences with Table 4, which shows the uniaxial compressive strength tests after 90 days.
Response 6:
I am sorry that my mistakes make you misunderstanding.
The test results after 540 days were used to test the strength of plastic concrete under uniaxial compression and true triaxial compression, and the test results of uniaxial compressive strength shown in Table 4 is cubes with a curing age of 540 days. The test results after 90 days was only used to test the elastic modulus of plastic concrete. The explanation is written in line 82-93:
Herein, 66 samples with a size of 150×150×150 mm3 and 33 samples with a size of 150×150×300 mm3 with 11 groups of mix proportions of plastic concrete were proportioned to study the constitutive model and failure criteria of plastic concrete. The mix proportions of plastic concrete specimens are reported in Table 1. Each group with a size of 150×150×150 mm3 and a curing age of 540 days had 6 samples that 3 samples were prepared to test the uniaxial compressive strength and 3 samples were proportioned to test true triaxial compressive strength. When the curing age is 540 days, the development of strength of plastic concrete tends to be stable, and the influence of curing age on strength can be ignored. There were 3 samples in each group with a size of 150×150×300 mm3 and a curing age of 90 days prepared to test the measured elastic modulus under uniaxial compressive stress. According to the mix proportions in Table 1, it can be seen from Table 1 that WB100 and C150, S05 and C120, B70 and CL180 are characterized by the same ingredients proportions. The data in Table 1 is arranged according to change of parameter.
The test results of a curing age of 540 days were used to establish the mathematical model of constitutive relationship of plastic concrete under true triaxial compression and compare the strength under uniaxial and true triaxial tests. So the fc* in Tables 7-9 is the uniaxial compressive strength of plastic concrete at an age of 540 days the strength I have written it in line 232-233. And the test results after 540 days were used in Eq(1), (4), (7).
f1, f2, are the confining pressure, f3 is true triaxial compressive strength.
Point 7.
In the last paragraph of p.2.2 you wrote when the results were rejected (15% deviation from the mean). Has this situation occurred and which models does it apply to?
Response 7:
It applies to the test of uniaxial compression. I have written is in line 137-139. It does not happen in our test, but it is the calculation method of uniaxial compressive strength of plastic concrete. So I think I have to explain it in article.
Point 8.
I am not a native speaker, but I can see some significant language mistakes. The text should be checked.
Response 8:
According to your suggestion, I read lots of paper and checked the article with my colleagues. I have corrected the mistakes of my article:
- I corrected tense in the article. In line 16, 17, 18, 20, 21,38, 39, 40,41, 43, 49,50, 51, 52, 53, 54, 55, 83, 86, 95,96, 99, 125, 130, 132, 135, 192, 302,304, 305, 306, 307, 309, 312, 313, 314, 317, 318, 319,320.
- I corrected the spelling mistakes of my article.
- I corrected the grammatical errors in the article.
- The unclear parts of the article have been revised.
- I analyzed the Table 4 in more depth.
- I added more pictures to make my conclusion clearer.
The main corrections focus on line10-21, 35-45, 66-79, 124-143, 147-154, 296-302.
Point 9.
Caption of Tables 7-9 - I think it's about Lode's angle, not Rhode
Response 9:
Thank you very much for correcting the mistakes in my article. I have replaced Rhode with Lode.
Point10.
Figure 2 may be slightly enlarged.
Response 10:
I have enlarged Figure 2.
Point 11.
Conclusions (there is more than one conclusion)
Response 11:
I refreshed the conclusion of my paper, added more percentages to paraphrase and change the tense into the past tense:
In this article, the mechanical performance of plastic concrete was tested and theoretically analyzed. Firstly, the results of the strength of plastic concrete under true triaxial and uniaxial compressive pressure were obtained through experiments. Secondly, according to results of the tests, the relationships between materials and confining pressure on strength of plastic concrete were analyzed. Thirdly, mathematic model of constitutive relationship and failure criteria based on strength of plastic concrete under true triaxial tests were established. The following conclusions can be obtained:
(1) According to the data shown in Table 4, the strength of plastic concrete under true triaxial compressive tests was greater than strength under uniaxial compressive tests due to lateral strain prevented by confining compressive stress. The triaxial compressive strength of plastic concrete increased by more than 70% compared with uniaxial compressive strength. The maximum growth rate could reach 181.85%
(2) With the increase of confining compressive stress, the lateral compressive stress had a better restraining effect on the transverse deformation of plastic concrete specimens, which delayed the appearance and development of micro-cracks in plastic concrete specimens. Therefore, the strength of plastic concrete increased with the confining compressive stress.
(3) Based on the experimental data and the existing constitutive model of concrete under triaxial compressive stress, a quartic polynomial constitutive model conforming to the constitutive characteristics of plastic concrete was established for the first time. Compared with the measured data, the ensemble average value of the ratio of σ3 /σ3f was 1.0037, the standard deviation was 0.0031 and the coefficient of variation was 0.0030, so the results of the mathematical model were in good agreement with the data obtained from the experiment.
(4) The failure criteria of plastic concrete under triaxial compressive stress in octahedral stress space was established. The general equation of the failure criteria was established according to the failure criteria. The average value of the ratio of calculated value to the experimental value was 1.002, the coefficient of variation was 0.0444, and the standard deviation was 0.0445 which proves that calculated value of failure criteria equation is in good agreement with the experimental value.

Round 2
Reviewer 3 Report
Dear authors.
Thanks for all the corrections. I have no more remarks. The manuscript may be published as is.
In my review, when I wrote about "Conclusions", I meant the title of the section. It must be plural because there are more than one conclusion. You did not understand me and developed conclusions. Of course they are correct, however, I still propose to change the title of the section 4 into "Conclusions".
This manuscript is a resubmission of an earlier submission. The following is a list of the peer review reports and author responses from that submission.
Round 1
Reviewer 1 Report
A well written paper with some issues to be addressed before being ready to be published:
- Page 2, line 21: there is “have the same mixture”, there should be “are characterized by the same ingredients proportions”.
- Page 2, line 25: there is “Type I-425 Portland cement” – a reference to a standard giving this cement class is needed.
- Page 2, Fig. 1: Granulometric properties of all used aggregates should be given in detail. Such parameters as “median diameter” and “fineness modulus” should be calculated and presented (with adequate references).
- Page 5, Table 4: Taking into account that only 3 specimens were used for each test calculating the values of mean square deviation and coefficient of variation is inappropriate. These values should be removed from the table.
- Page 7 – 10, Fig. 2 – 12. All relations should be presented using the same length of vertical and horizontal axis (e.g. 16 and 7 respectively). Then the comparison and analysis of the presented results would be enabled.
Reviewer 2 Report
The reviewer thanks the authors for preparing the manuscript. However, the paper has some problems.
- This paper is just repeated a previous paper in an international conference. Almost results used in this paper are adopted from the previous one, and the figures are also the same.
- The experimental works of this paper have some problems. For example, " According to the mix proportions in Table 1, it can be seen from Table 1 that WB100 and CL150, S05 and CL120, B70 and CL180 have the same mixture", the reviewer did not know the purpose of these same mixtures.
- The specimens were cured for 90 days and 540 days, each mix proportion had 6 specimens, 3 specimens were used for true triaxial compressive stress, and others used for unxiaxial compressive stress, but the total number of specimens is 66 specimens.
- The paper was not prepared carefully with a lost of mistakes and poor English.
As a result, the reviewer would like to reject this paper and hope the authors revise it and prepare it carefully.
Regards,
Reviewer 3 Report
The topic of the paper is interesting and suits the Journal of Materials. However, minor revision is required before this manuscript is qualified to be published in this prestigious journal. The manuscript is needed to be revised grammatically. The authors are required to check the whole manuscript with a grammar specialist as it has several grammatical errors. Only after revising the manuscript based on the comments, the paper is suggested to be published in JM. Further information on various issues identified in the manuscript appears below:
- The authors have done a great job on the literature review. Please add more literature with regards to the works that have been published in the Journal of Materials.
- More information on triaxial test on concrete can be found here:
Shannag, M. J. "High strength concrete containing natural pozzolan and silica fume." Cement and concrete composites 22.6 (2000): 399-406.
Ramezanianpour, A. A., et al. “Effect of combined carbonation and chloride ion ingress by an accelerated test method on microscopic and mechanical properties of concrete.” Construction and Building Materials (2014): 58, 138-146.
and,
Ghahari, S. et al. "Performance assessment of natural pozzolan roller compacted concrete pavements." Case studies in construction materials (2017): 82-90.
- Avoid all self-citations as it seems that all self-citations are unnecessary in this paper.
- Please provide the exact specification of the materials and the strengths.
- Please provide a more detailed reasoning behind the structure behavior. The details should include the rigid numbers or percentages.
- Please indicate how many samples for each experiment have been used. Please revise the other experiments respectively.
- Please describe the process of each experiment. Also indicate the model of each tool that is used in the experiment. What is the accuracy of each machine? Please explain them accurately.
- Please add error bars to the figures where feasible.
- Conclusion needs to be written in paragraphs. Please illustrate the main conclusions in the manuscript. Please paraphrase your results and discussions and use them in the conclusion part.
Round 2
Reviewer 2 Report
Dear authors,
Thank you very much.
After my comments, the authors have revised the paper and it became a new submission as I wrote last time.
The reviewer would like to keep the original idea because many parts of this paper have been published in the international conference. Thus, this paper must be different from that one in order to be accepted for publication.
Regards,